# Evaluation of the Usability of UAV LiDAR for Analysis of Karst (Doline) Terrain Morphology

**DOI:** 10.3390/s24217062

**Published:** 2024-11-01

**Authors:** Juneseok Kim, Ilyoung Hong

**Affiliations:** Department of Drone & GIS Engineering, University of Namseoul, Cheonan City 31020, Republic of Korea; junesoek.kim@nsu.ac.kr

**Keywords:** UAV, LiDAR, doline, DEM

## Abstract

Traditional terrain analysis has relied on Digital Topographic Maps produced by national agencies and Digital Elevation Models (DEMs) created using Airborne LiDAR. However, these methods have significant drawbacks, including the difficulty in acquiring data at the desired time and precision, as well as high costs. Recently, advancements and miniaturization in LiDAR technology have enabled its integration with Unmanned Aerial Vehicles (UAVs), allowing for the collection of highly precise terrain data. This approach combines the advantages of conventional UAV photogrammetry with the flexibility of obtaining data at specific times and locations, facilitating a wider range of studies. Despite these advancements, the application of UAV LiDAR in terrain analysis remains underexplored. This study aims to assess the utility of UAV LiDAR for terrain analysis by focusing on the doline features within karst landscapes. In this study, we analyzed doline terrain using three types of data: 1:5000 scale digital topographic maps provided by the National Geographic Information Institute (NGII) of Korea, Digital Surface Models (DSMs) obtained through UAV photogrammetry, and DEMs acquired via UAV LiDAR surveys. The analysis results indicated that UAV LiDAR provided the most precise three-dimensional spatial information for the entire study site, yielding the most detailed analysis outcomes. These findings suggest that UAV LiDAR can be utilized to represent terrain features with greater precision in the future; this is expected to be highly useful not only for generating contours but also for conducting more detailed topographic analyses, such as calculating the area and slope of the study sites.

## 1. Introduction

In South Korea, traditional terrain analysis has relied heavily on 1:5000 scale digital topographic maps produced by the National Geographic Information Institute (NGII) or publicly available Digital Elevation Models (DEMs). DEM refers to a digital elevation model created in a grid format using digital terrain data or triangulated irregular networks (TINs), representing the elevation of the Earth’s surface while excluding surface cover such as buildings, vegetation, and artificial structures. The DEM is a crucial terrain product and a fundamental requirement in many terrain analysis applications. However, recent advances have led to the adoption of Unmanned Aerial Vehicles (UAVs) for terrain analysis in small areas. UAVs offer faster and more accurate location data acquisition compared to manned aircraft, achieving location accuracy within ±5 cm, which meets the legal standards for detailed terrain surveys. Alongside UAVs, LiDAR technology has emerged as a powerful tool in spatial information science, offering high-resolution three-dimensional terrain data. The integration of UAVs with LiDAR equipment combines the precision of traditional aerial LiDAR with the flexibility of UAVs, enhancing the efficiency and accuracy of terrain analysis.

Karst landscapes, such as dolines (sinkholes), are of particular interest due to their unique geological features. Dolines are important for agriculture, biodiversity, and conservation efforts, as seen in areas like the Sobaek Mountain Range and Samcheok in Gangwon Province, South Korea. Accurate mapping of dolines is essential for understanding these landscapes, and UAV LiDAR presents an opportunity to improve doline detection and analysis through high-resolution data acquisition.

The utility of LiDAR in terrain analysis has been demonstrated in various studies and Table 1 summarizes the related works. These studies highlight the effectiveness of UAVs and LiDAR for terrain analysis. However, there is limited research that combines UAV LiDAR specifically for doline detection and analysis.

This study aims to fill the gap in research by evaluating the effectiveness of UAV LiDAR in doline detection and terrain feature analysis. The specific contributions of this study are the following: The first is the acquisition of high-resolution UAV LiDAR data in karst landscapes; the second is the processing and analysis of the data to detect dolines; the third is the comparison of UAV LiDAR with traditional methods for accuracy and cost-efficiency; the fourth is the development of a methodology for detailed terrain feature analysis using UAV LiDAR.

The remainder of the paper is structured as follows: Section 2 outlines the methodology, including the UAV LiDAR data acquisition process and analysis techniques. Section 3 presents the results, focusing on the detection of dolines and their accuracy compared to traditional methods. Section 4 discusses the findings and implications for future terrain analysis. Section 5 concludes the study, summarizing the key contributions and suggesting areas for further research.

## 2. Materials and Methods

### 2.1. Study Site

Danyang (Figure 1), located in South Korea, is geographically bordered by the Baekdudaegan mountain range to the east and south, with the Namhan River running through its center. The northern region of Danyang is characterized by limestone formations, creating a karst landscape with features such as karren, dolines, and numerous lime caves. In karst landscapes, limestone dissolution by groundwater leads to the development of lime caves, while surface dissolution by rainwater results in the formation of dolines. Danyang’s geological structures, including faults and folds, add significant geological value to the area.

### 2.2. LiDAR Point Cloud Acquisition Strategy

The accuracy of LiDAR surveys depends not only on the payload’s return mode and scanning mode settings but also on factors such as the UAV’s flight direction, flight speed, and the characteristics of the terrain. A thorough preliminary review is necessary. Based on an assessment of the terrain elevation in the study site, which revealed an elevation difference of approximately 50 to 70 m (Figure 2), it was classified as a hilly area. Consequently, the flight parameters were set as follows: an altitude of 100 m, a flight speed of 7.5 m/s, a side overlap of 50%, and a camera angle of –90°.

The return mode was set to the penta returns mode. In this mode, the first return provides height information, the intermediate returns capture the structure of objects, and the last return provides ground information. The penta returns mode is suitable for areas with dense vegetation due to its stronger laser pulse penetration. The point cloud rate, also known as the sampling rate or pulse frequency, indicates the maximum number of laser beams emitted per unit time. Higher frequencies result in more measurement points and higher accuracy under the same conditions. The field of view (FOV), also known as the scan angle, represents the angle of the laser beams during scanning. In the repeat scanning mode, the FOV is 70° × 30°. This mode, although having a narrower scan range, offers higher accuracy and is recommended for high-precision mapping. Table 2 shows the setting of LiDar payload parameters.

### 2.3. Research Matrials

The data used in this study include the most recent 1:5000 scale digital topographic map produced in 2023, obtained from the National Geographic Information Institute of Korea. Additionally, LiDAR data and optical images necessary for the research were acquired directly through UAV (Unmanned Aerial Vehicle) imaging. Figure 3 illustrates the flow of the study.

### 2.4. Aerial Photography Equipment

The UAV-based mobile mapping system used in this study is the DJI Matrice 300 RTK with a Zenmuse L2 LiDAR sensor, capable of collecting both LiDAR and imagery data. Table 3 presents a UAV-based mobile mapping system.

### 2.5. Flight Plan

The flight area for the study site was defined, and parameters such as flight altitude, duration, and coverage area were verified (Figure 4). During aerial photography in the flight area, Real Time Kinematics (RTKs) surveying was conducted without using Ground Control Points (GCPs). The RTK positioning accuracy of the Matrice 300 is 1 cm + 1 ppm horizontally and 1.5 cm + 1 ppm vertically. The Virtual Reference System (VRS) creates a virtual reference point through network modeling, simulating observations from an arbitrary nearby reference station, and determines precise mobile station positions using RTKs with this virtual reference point. RTK-based unmanned aerial photography involves the UAV’s GNSS receiver being connected via WiFi or unmanned mobile communication networks to a base station or VRS, transmitting calibration data for real-time determination of the UAV’s position and attitude using relative positioning techniques.

When connecting the UAV to the Network RTK service provided by the National Geographic Information Institute (NGII) of South Korea, the RTK signal continuously outputs a fixed solution throughout the flight. The station file is automatically saved to the Zenmuse L2′s result file. Horizontal datum settings were configured as EPSG: 5187, Korea 2000/East Belt 2010, and the Geoid settings were set to the Korean National Geoid Model (KNGeoid 18).

### 2.6. Digital Elevation Model Data Processing

In conventional airborne laser scanning, the pixel size of DEM grids is typically 0.5 m, and the point density criterion is set at over 50 points per square meter. Horizontal position accuracy is maintained at a flight altitude divided by 1000, while vertical position accuracy ensures a maximum error within 75 cm and RMSE within 50 cm. The density of the point cloud obtained in this study is 197/m^2^, the DEM was created from ground data obtained via UAV LiDAR, classified into ground and non-ground points. The pixel size of the DEM was set to 0.25 m. Table 4 shows the flight configuration for LiDAR canning and Figure 5 shows the DEM acquired with UAV LiDAR.

### 2.7. Orthophoto and Digital Surface Model Data Processing

Orthophotos and Digital Surface Models (DSMs) were generated automatically using UAV aerial photographs and Exterior Orientation (EO) parameters in software. Table 5 shows the flight configuration for photogrammetry and Figure 6 is the result of image-based mapping.

The pixel size of the DEM, created from LiDAR point cloud data, is 0.25 m. For the DEM based on contour data from the 1:5000 scale digital topographic map with a contour interval of 5 m, the grid spacing was set to 5 m. Figure 7 shows the three types of data processing results.

## 3. Results

### 3.1. Elevation Profile Analysis

We conducted a comparative analysis of the accuracy of three datasets: the LiDAR-based DEM, the image-based DSM, and the DEM based on digital topographic maps, using elevation profiles derived from the processed data. Firstly, we selected four regions for analysis, including areas with sinkholes and some vegetation cover (Figure 8). To enhance the three-dimensional visual effect and depict changes in shading, we created Shaded Relief Maps (SRMs) based on raster elevation data.

Using the SRMs as a reference, we designated cross-section lines passing through sinkholes and vegetation areas in each experimental region (Figure 9). We compared elevation profiles accordingly (Figure 10). In regions A, B, and D, both LiDAR-based DEMs and image-based DSMs clearly depicted altitude changes at sinkholes, exhibiting generally similar elevation profile shapes. Upon closer inspection, the elevation profiles from the image-based DSM appeared slightly more rugged. The DEM based on digital topographic maps did not detect all sinkholes across the entire area and showed gentle elevation profile shapes without significant altitude changes. When we examined the elevation profile of region C, the image-based DSM displayed a dip in the elevation profile, suggesting a detection of sinkholes. This dip is attributed to interference from vegetation, resulting in pixels with elevation values of zero, as indicated by the black areas in the SRMs.

### 3.2. Doline Detection Comparison

For detecting doline depressions, previous methods have utilized elevation differences in DEMs [28], while recent studies have explored deep learning approaches [29]. In this study, we employed a Python package proposed by [30] for depression detection. Figure 11 below presents a comparative illustration of doline detection results. Black indicates detections specifically classified as doline outlines, while gray represents detections of general depressions. In comparing doline detection between LiDAR-based DEMs and image-based DSMs, the overall positions and shapes of dolines in the study site were similar. However, in some locations, the LiDAR-based DEM identified a single doline where the image-based DSM detected two separate dolines, or instances where discrepancies occurred with some dolines being missed or additional ones being identified between the datasets.

Upon executing the doline extraction algorithm on the image-based DSM, a total of 21 depressions were identified. Conversely, the LiDAR-based DEM yielded 13 depressions, while the DEM obtained from digital topographic maps did not extract any. The exploration program allows adjustment of cell size and depth settings; for this study, a depth value of 1 m and a pixel value of 1000 were set to detect depressions. When we compared the results between image-based DSMs and LiDAR-based DEMs, the image-based DSMs detected 9 out of 21 depressions that were not dolines, while the LiDAR-based DEMs had 3 erroneous detections out of 13. The image-based DSM detected relatively more depressions overall, but it also exhibited a higher error rate in doline detection compared to the LiDAR-based DEM. This outcome suggests that the errors in the image-based DSM results were primarily due to the detection of depressions caused by vegetation interference.

## 4. Discussion

This study demonstrated the strengths of UAV LiDAR systems in analyzing the morphology of doline terrains compared to traditional research methods. The research focused on comparing the quality of terrain data produced by UAV LiDAR, photogrammetry technology, and digital topographic maps. Through a case study analyzing the morphology of doline terrains in Danyang, Chungbuk, in South Korea, where limestone formations are prominent, the strengths and limitations of UAV LiDAR were evaluated as follows.

Firstly, the evaluation concerns data quality. Initially, the 1:5000 scale digital topographic map produced DEMs that provided the coarsest level of terrain detail. Next, point clouds extracted using photogrammetry primarily covered the surface, with lower point density being observed in densely vegetated areas. This aligns with previous research indicating that image-based methods yield relatively poorer results in regions with low texture. In the case of Danyang, LiDAR point clouds covered larger areas and offered higher point density compared to image-based point clouds. LiDAR point cloud data could capture points beneath vegetation, enabling more accurate DEMs in complex terrains. Analysis of cross-section lines passing through sinkholes and vegetation areas revealed subtle differences in quality between the three datasets.

Secondly, the evaluation focuses on quantitative analysis of doline terrains. Using a doline extraction program, the DEM created from the 1:5000 scale digital topographic map detected zero dolines. In contrast, using image-based DSMs, a total of 21 dolines were extracted, while LiDAR-based DEMs identified 13 dolines under the same conditions. In the case of image-based DSMs, more dolines were erroneously identified due to depressions caused by vegetation interference being mistaken for dolines. The DEMs generated using LiDAR provided detailed data sufficient to identify sinkholes within dolines.

Thirdly, the advantage of UAV LiDAR technology is highlighted. Point cloud data acquired through LiDAR boasts exceptionally high spatial resolution. UAV offer the advantage of acquiring accurate surface data even in challenging terrains like mountainous regions and densely vegetated areas. Compared to satellite or airborne LiDAR data, UAV LiDAR provides higher resolution and is less affected by cloud cover during data collection. Additionally, UAV equipped with optical cameras can capture high-resolution aerial RGB images, which are more cost-effective and easier to process compared to using LiDAR sensors alone. Photogrammetry technology excels in general terrain surveys but has limitations in surface analysis due to variations in vegetation type and density. While LiDAR sensors can be mounted on aircraft, the associated costs and maintenance as well as the need to plan data collection around weather conditions pose significant barriers to widespread use. In contrast, UAV LiDAR systems are more cost-effective, accessible, and easier to plan for data acquisition.

## 5. Conclusions

This study highlighted the benefits of using UAV LiDAR systems for the analysis and detection of doline landforms. It compared three methods—UAV LiDAR-based DEMs, image-based DSMs, and digital topographic map-based DEMs—in terms of quality, accuracy, and spatial coverage. The comparison revealed the distinct strengths and weaknesses of each approach, emphasizing the effectiveness of UAV LiDAR in geomorphic environments.

UAV LiDAR proved to be especially valuable due to its ability to capture high-resolution data with high point density and broad spatial coverage, enabling more reliable DEM generation in complex terrains. This was demonstrated through a case study of doline landforms in Danyang. First, UAV LiDAR’s ability to penetrate vegetation allowed for accurate terrain mapping even in densely forested areas. Second, while image-based DSMs are effective at capturing surface details, their point density decreases in areas with low texture, reducing accuracy. Third, quantitative differences in elevation data from the three methods were identified using a doline detection program, underscoring the advantages of UAV LiDAR for small-scale terrain analysis, such as dolines in Korea.

Future research should expand the investigation into the use of UAV LiDAR and image-based technologies across various terrain types. Integrating these technologies could yield more precise and comprehensive terrain models, enhancing the accuracy of detailed landform analysis. Furthermore, the development of automated data processing and analysis methods will be key to improving the efficiency of detecting specialized landforms such as dolines.

In conclusion, this study demonstrates that UAV LiDAR and image-based technologies are essential tools for analyzing and monitoring doline landforms. By evaluating their strengths and limitations, it is evident that future advancements in and integration of these technologies can significantly enhance the precision and efficiency of detailed landform analysis, particularly in challenging terrain environments.

## Figures and Tables

**Figure 1 sensors-24-07062-f001:**
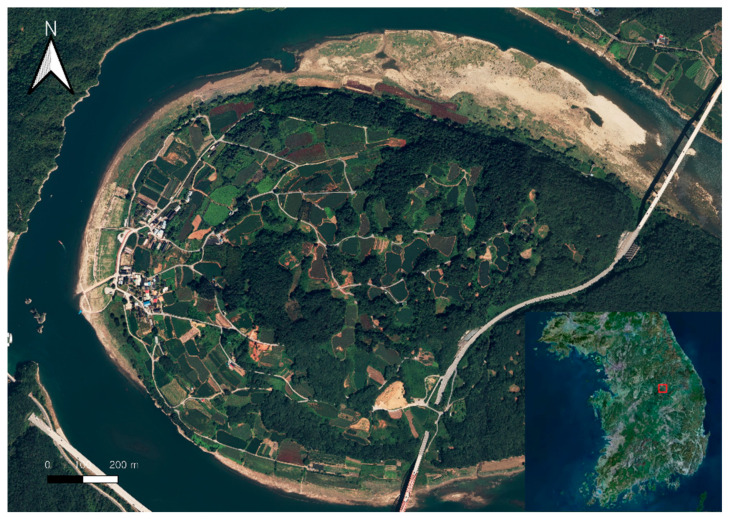
The study site, Danyang, for doline topography (The red square indicates the location of Danyaning in the Republic of Korea).

**Figure 2 sensors-24-07062-f002:**
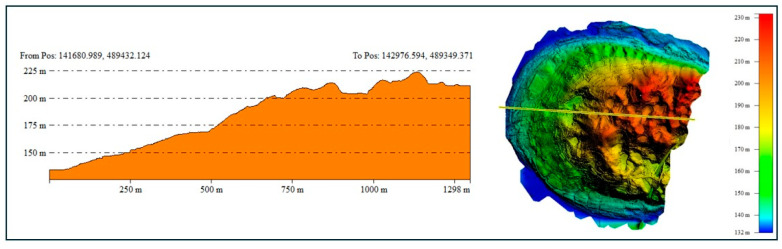
Elevation profile of the study site.

**Figure 3 sensors-24-07062-f003:**
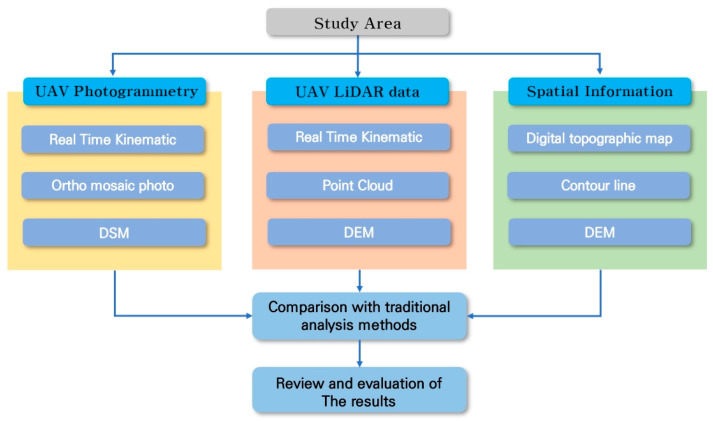
Research flow chart.

**Figure 4 sensors-24-07062-f004:**
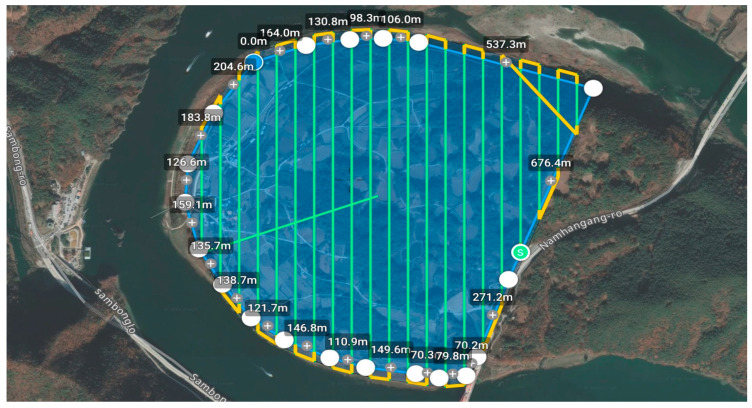
Flight route of UAV over study site (DJI RC Plus controller screen).

**Figure 5 sensors-24-07062-f005:**
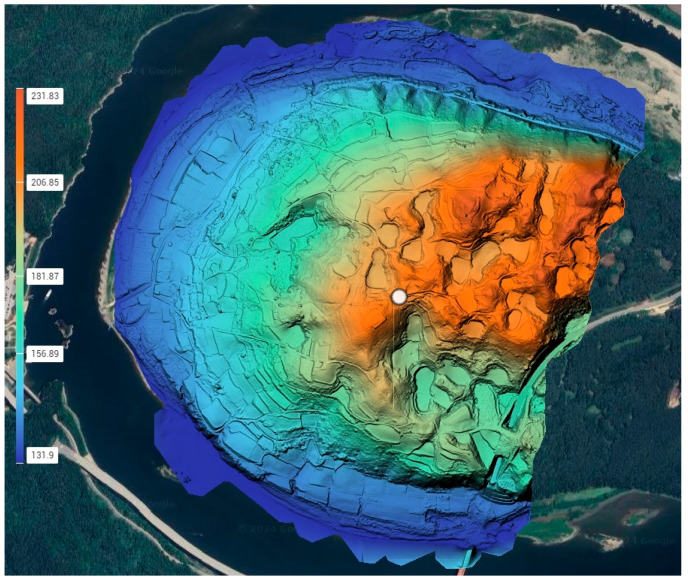
Digital elevation model acquired with UAV LiDAR. (The result of processing in DJI Terra software).

**Figure 6 sensors-24-07062-f006:**
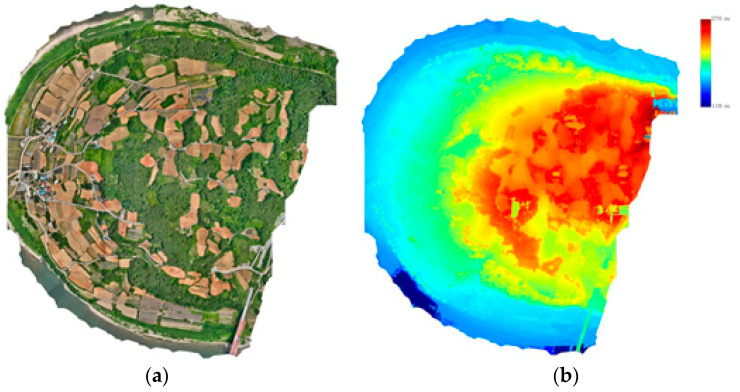
Image-based mapping: (**a**) Orthophoto; (**b**) Digital surface model. (The result of processing in PIX4D mapper software).

**Figure 7 sensors-24-07062-f007:**
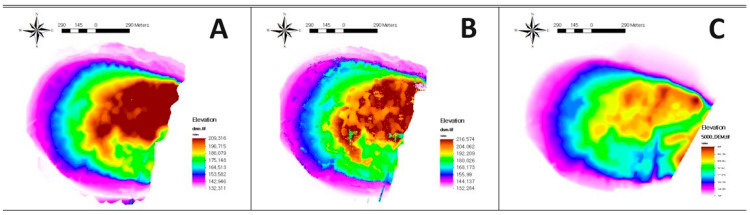
Types of data processing results: (**A**) UAV LiDAR-based DEM; (**B**) UAV Images-based DSM; (**C**) Contour-based DEM of 1/5000 scale digital topographic map.

**Figure 8 sensors-24-07062-f008:**
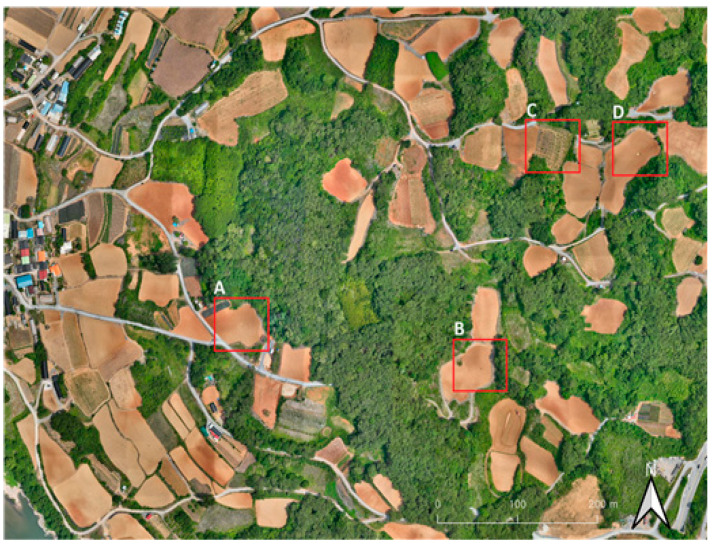
Experimental areas for elevation profile analysis A, B, C, and D.

**Figure 9 sensors-24-07062-f009:**
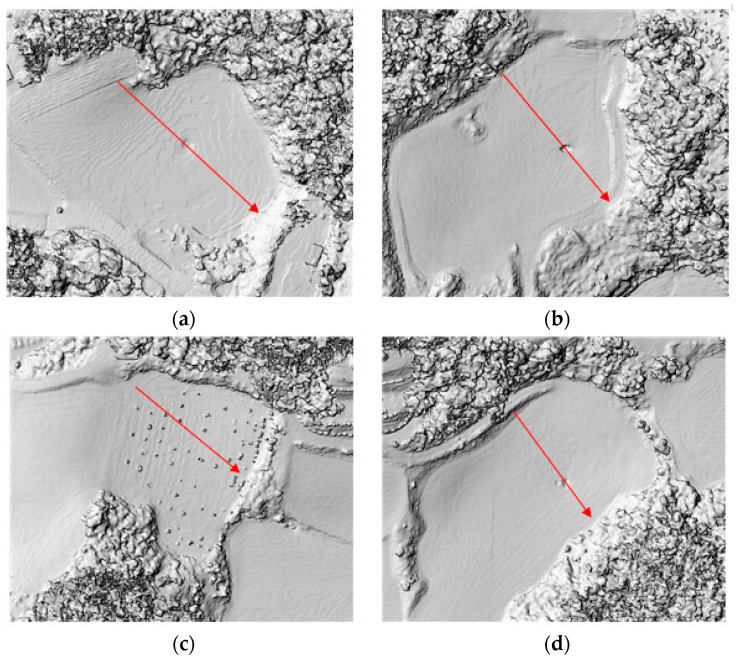
Cross-section lines (red arrows) for experimental areas: (**a**) Cross-section line of area A; (**b**) Cross-section line of area B; (**c**) Cross-section line of area C; (**d**) Cross-section line of area D.

**Figure 10 sensors-24-07062-f010:**
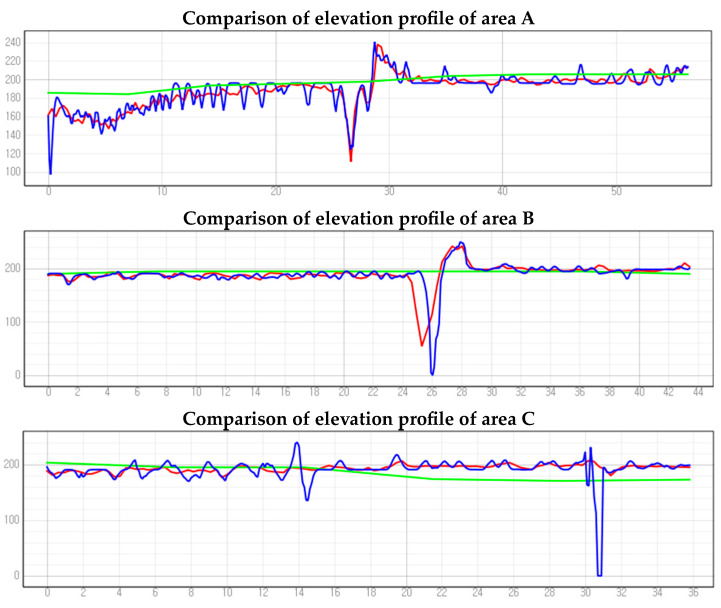
Comparison of elevation profile A, B, C, and D areas.

**Figure 11 sensors-24-07062-f011:**
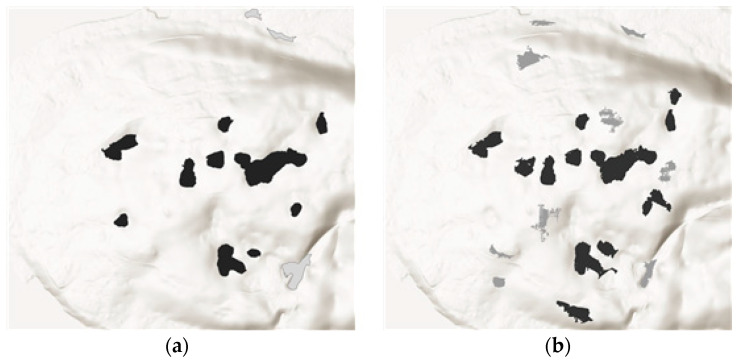
Comparison of doline delineations on two data types: (**a**) UAV LiDAR DEM. (**b**) DSM from photogrammetry and black color: true doline, gray color: false doline.

**Table 1 sensors-24-07062-t001:** Related Works.

Reference #	Focus	Methods/Tools Used
[1,2,3,4,5,6,7]	UAV-based terrain analysis	UAV photogrammetry, DEM
[8,9,10,11,12]	LiDAR technology in spatial information	Aerial LiDAR, UAV LiDAR
[13,14,15,16]	Dolines detection using LiDAR in karst landscapes	Airborne LiDAR, DEM
[17,18]	Doline detection	GIS, DEM, multi-layer depth maps
[19,20,21]	UAV LiDAR in terrain feature analysis and road geometry	UAV LiDAR, DTM, deep learning
[22,23]	Automated detection and mapping of sinkholes	High-resolution maps, digital data
[24,25,26,27]	UAV for cost-effective high-resolution terrain analysis	UAV, RGB imagery

**Table 2 sensors-24-07062-t002:** LiDAR payload parameters settings.

Category	Parameter
Return mode	Penta returns
Sampling rate	240 kHz
Scanning mode	Repetitive
Altitude	80 m~150 m
Side overlap	50%
Camera angle	−90

**Table 3 sensors-24-07062-t003:** UAV-based mobile mapping system.

Matrice 300 RTK	Zenmuse L2
RTK accuracy H: 0.1 m V: 0.15 m	Ranging accuracy 2 cm@150 m
Hovering accuracy H:0.1 m V:0.1 m	Max returns supported 5
Maximum flight time 55 min	Min detection range 3 m
GPS, GLONASS, BeiDou, Galileo	Laser spot size H:4 cm V:12 cm
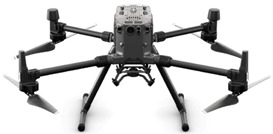	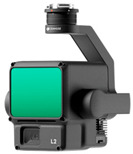

**Table 4 sensors-24-07062-t004:** Flight configuration for LiDAR scanning in study site.

Category	Parameter
Altitude	Horizontal flight 100 m
Speed	7.5 m/s
Scanning mode	Repetitive scanning
Side overlap ratio	50%
Camera angle	−90

**Table 5 sensors-24-07062-t005:** Flight configuration for photogrammetry in study site.

Category	Parameter
Date	10 June 2024
Image quantity	830
Average flight altitude	131.09 m
Area covered	1.012 km^2^
Average GSD	3.92 cm/pix

## Data Availability

Upon a reasonable request from the corresponding author.

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
