# Peer review of "Evaluation of the Usability of UAV LiDAR for Analysis of Karst (Doline) Terrain Morphology"

_sensors, 2024, doi:10.3390/s24217062_

Round 1

Reviewer 1 Report

Comments and Suggestions for Authors

The article highlights the advantages of UAV-integrated LiDAR for more flexible and accurate terrain data collection, demonstrating its precision in analyzing doline features and its potential for detailed topographic studies. While the study's results are intresting, herein are  some suggestions for improving the presentation: -The introduction did not highlight the study's motivation clearly. It's recommended to divide the introduction into four subsections: (1.1) Background and Motivation, (1.2) Related Works (summarized in a table), (1.3) Contributions, and (1.4) Paper Organization.-Improve the resolution of the figures.-The discussion section is brief and should be expanded for deeper analysis.

Comments on the Quality of English Language

OK

Author Response

comment

"The article highlights the advantages of UAV-integrated LiDAR for more flexible and accurate terrain data collection, demonstrating its precision in analyzing doline features and its potential for detailed topographic studies. While the study's results are intresting, herein are  some suggestions for improving the presentation: -The introduction did not highlight the study's motivation clearly. It's recommended to divide the introduction into four subsections: (1.1) Background and Motivation, (1.2) Related Works (summarized in a table), (1.3) Contributions, and (1.4) Paper Organization.-Improve the resolution of the figures.-The discussion section is brief and should be expanded for deeper analysis."

response

We changed the introduction according to reviewer's suggesetion and enhaced the qualtiy of figues and updated the discussion parts which can be found at new submitted  paper file.

Here is the modified introduction;

  1. Introduction

In South Korea, traditional terrain analysis has heavily relied on 1:5,000 scale digital topographic maps produced by the National Geographic Information Institute (NGII) or publicly available Digital Elevation Models (DEMs). DEM refers to a digital elevation model created in a grid format using digital terrain data or triangulated irregular networks (TIN), representing the elevation of the Earth's surface while excluding surface cover such as buildings, vegetation, and artificial structures. DEM is a crucial terrain product and a fundamental requirement in many terrain analysis applications. However, recent advances have led to the adoption of Unmanned Aerial Vehicles (UAV) for terrain analysis in small areas. UAVs offer faster and more accurate location data acquisition compared to manned aircraft, achieving location accuracy within ±5 cm, which meets the legal standards for detailed terrain surveys. Alongside UAVs, LiDAR technology has emerged as a powerful tool in spatial information science, offering high-resolution three-dimensional terrain data. The integration of UAVs with LiDAR equipment combines the precision of traditional aerial LiDAR with the flexibility of UAVs, enhancing the efficiency and accuracy of terrain analysis.

Karst landscapes, such as dolines (sinkholes), are of particular interest due to their unique geological features. Dolines are important for agriculture, biodiversity, and conservation efforts, as seen in areas like the Sobaek Mountain Range and Samcheok in Gangwon Province, South Korea. Accurate mapping of dolines is essential for understanding these landscapes, and UAV LiDAR presents an opportunity to improve doline detection and analysis through high-resolution data acquisition.

In South Korea, traditional terrain analysis has heavily relied on 1:5,000 scale digital topographic maps produced by the National Geographic Information Institute (NGII) or publicly available Digital Elevation Models (DEMs). DEM refers to a digital elevation model created in a grid format using digital terrain data or triangulated irregular networks (TIN), representing the elevation of the Earth's surface while excluding surface cover such as buildings, vegetation, and artificial structures. DEM is a crucial terrain product and a fundamental requirement in many terrain analysis applications. However, recent advances have led to the adoption of Unmanned Aerial Vehicles (UAV) for terrain analysis in small areas. UAVs offer faster and more accurate location data acquisition compared to manned aircraft, achieving location accuracy within ±5 cm, which meets the legal standards for detailed terrain surveys. Alongside UAVs, LiDAR technology has emerged as a powerful tool in spatial information science, offering high-resolution three-dimensional terrain data. The integration of UAVs with LiDAR equipment combines the precision of traditional aerial LiDAR with the flexibility of UAVs, enhancing the efficiency and accuracy of terrain analysis.

Karst landscapes, such as dolines (sinkholes), are of particular interest due to their unique geological features. Dolines are important for agriculture, biodiversity, and conservation efforts, as seen in areas like the Sobaek Mountain Range and Samcheok in Gangwon Province, South Korea. Accurate mapping of dolines is essential for understanding these landscapes, and UAV LiDAR presents an opportunity to improve doline detection and analysis through high-resolution data acquisition.

The utility of LiDAR in terrain analysis has been demonstrated in various studies and Table 1 summaries the related works. These studies highlight the effectiveness of UAVs and LiDAR for terrain analysis. However, there is limited research that combines UAV LiDAR specifically for doline detection and analysis.

Table 1. Related Works

Reference #

Focus

Methods/Tools Used

[1-4, 28,29, 30, 31]

UAV-based terrain analysis

UAV photogrammetry, DEM

[5-8]

LiDAR technology in spatial information

Aerial LiDAR, UAV LiDAR

[10-12, 25]

Dolines detection using LiDAR in karst landscapes

Airborne LiDAR, DEM

[13] [26]

Doline detection  

GIS, DEM, multi-layer depth maps

[14-16]

UAV LiDAR in terrain feature analysis and road geometry

UAV LiDAR, DTM, deep learning

[17,18]

Automated detection and mapping of sinkholes

High-resolution maps, digital data

[19-21,27]

UAV for cost-effective high-resolution terrain analysis

UAV, RGB imagery

This study aims to fill the gap in research by evaluating the effectiveness of UAV LiDAR in doline detection and terrain feature analysis. The specific contributions of this study are like followings. First is the acquisition of high-resolution UAV LiDAR data in karst landscapes. Second is the Processing and analysis of the data to detect dolines. Third is the comparison of UAV LiDAR with traditional methods for accuracy and cost-efficiency. Fourth is the development of a methodology for detailed terrain feature analysis using UAV LiDAR.

The remainder of the paper is structured as follows: Section 2 outlines the methodology, including the UAV LiDAR data acquisition process and analysis techniques. Section 3 presents the results, focusing on the detection of dolines and their accuracy compared to traditional methods. Section 4 discusses the findings and implications for future terrain analysis. Section 5 concludes the study, summarizing the key contributions and suggesting areas for further research.

Reviewer 2 Report

Comments and Suggestions for Authors

1. Introduction is not good enough, for example I know  LiDAR, but not DEM and DSM, all three ways should be explained more to show authors' understanding and the development of those.

2. Figure 3, the flow is not typical, should be more rigorous.

3. more references need.

overall, this is more like a practice than an academic research, some theory details could help.

Comments on the Quality of English Language

It is readable and understandable.

Author Response

Comment 1. Introduction is not good enough, for example I know  LiDAR, but not DEM and DSM, all three ways should be explained more to show authors' understanding and the development of those.

Response 1: We added the additonal explanation as like following

In traditional terrain analysis within South Korea, the 1:5,000 scale digital topographic maps produced by the National Geographic Information Institute (NGII) or publicly available DEM have been primarily utilized.  DEM refers to a digital elevation model created in a grid format using digital terrain data or triangulated irregular networks (TIN), representing the elevation of the Earth's surface while excluding surface cover such as buildings, vegetation, and artificial structures. DEM is a crucial terrain product and a fundamental requirement in many terrain analysis applications. Recently, there has been a surge in research applying UAV for terrain analysis in small areas, leveraging their advantages of faster and more accurate acquisition of location data compared to manned aircraft[1–4].

Comment 2. Figure 3, the flow is not typical, should be more rigorous.

Response 2. we recreate figure 3, Reserch flow chart.

Commnet 3. more references need.

Response 3. we added more related references from Sensors, Remte Sensing, Applied Science

Reviewer 3 Report

Comments and Suggestions for Authors

I found no substantial innovations in the manuscript, and some of the findings were obvious. In addition, the quality of the figures is very poor, and the author should spend more effort to improve the quality of the manuscript from all aspects.

Blank space for "Karst" in the title of the manuscript;

Kewords: doline→Doline

Remove the text from Figure 1 and place it in the title

Figure 2: Since you've already used drones to get terrain data, why not use drone images and terrain presentations? Using Google Earth satellite images and topographic profiles of the Earth and showing them to the reader is extremely unscrupulous

Number before check box text in Figure 3, 1,2? or 1,3?

Author Response

Comment 1: Blank space for "Karst" in the title of the manuscript;

Response 1: we corrected

Comment 2 : Kewords: doline→Doline

Response 2: we corrected

Comment 3. Remove the text from Figure 1 and place it in the title

Response 2: we corrected

comment 4:  Figure 2: Since you've already used drones to get terrain data, why not use drone images and terrain presentations? Using Google Earth satellite images and topographic profiles of the Earth and showing them to the reader is extremely unscrupulous

Response 4, we recreated figure 2 

Please see the attached file. The author's response.

Reviewer 4 Report

Comments and Suggestions for Authors

This paper investigates the usability of UAV LIDAR in doline terrain analysis and analyzes funnel terrain using three types of data, 1:5000 scale digital topographic maps provided by the National Geographic Information Institute (NGII) of Korea, Digital Surface Models (DSM) obtained through UAV photogrammetry, and DEM acquired via UAV LiDAR surveys. The results show that UAV LIDAR achieves better results.

However, to improve the quality of this paper in the future, the following suggestions should be taken into account.

1.     It is suggested to briefly explain the reasons for choosing Danyang as the study site among many sites in Section 2.1.

2.     What do the sequence numbers 1 and 3 (‘1. 2D Ortho Mosaic’ and ‘3. LiDAR Point Cloud’) mean in the flow chart shown in Figure 3? In addition, the image in Figure 3 is fuzzy, so it is recommended to replace it with a high-resolution image.

3.     Figure 10 shows the three comparison curves. Is there a standard curve for reference? If so, please add it.

4.     The data in Figure 7 is blurry, it is recommended to replace it with a clearer picture.

5.     It is suggested to refine the content of the conclusion.

Comments on the Quality of English Language

Moderate editing of English language required.

Author Response

comment 1. It is suggested to briefly explain the reasons for choosing Danyang as the study site among many sites in Section 2.1.

respopnse 1: we correct section 2.1

comment 2. What do the sequence numbers 1 and 3 (‘1. 2D Ortho Mosaic’ and ‘3. LiDAR Point Cloud’) mean in the flow chart shown in Figure 3? In addition, the image in Figure 3 is fuzzy, so it is recommended to replace it with a high-resolution image.

response 3. we recreated figure 3

comment 3 Figure 10 shows the three comparison curves. Is there a standard curve for reference? If so, please add it.

response 4: There are no separate standard curves because they are height values for DEMs in areas A, B, C, and D. modify

comment 5. The data in Figure 7 is blurry, it is recommended to replace it with a clearer picture.

response 5: we made the highest resolution

comment 6 It is suggested to refine the content of the conclusion.

respose 6. we refined the conclusion

Round 2

Reviewer 3 Report

Comments and Suggestions for Authors

The author has revised the manuscript according to the revision suggestions, and I think the manuscript is acceptable for now.

Reviewer 4 Report

Comments and Suggestions for Authors

Accept in present form

Comments on the Quality of English Language

Moderate editing of English language required